# CAR T Cell Therapy in Glioblastoma: Overcoming Challenges Related to Antigen Expression

**DOI:** 10.3390/cancers15051414

**Published:** 2023-02-23

**Authors:** Andrew S. Luksik, Eli Yazigi, Pavan Shah, Christopher M. Jackson

**Affiliations:** Department of Neurosurgery, Johns Hopkins University School of Medicine, Baltimore, MD 21205, USA

**Keywords:** immunotherapy, glioblastoma, CAR T

## Abstract

**Simple Summary:**

Glioblastoma is the most common brain cancer, and prognosis remains dismal. Continued efforts to find better treatments have shown the promise of immunotherapy, whereby the immune response against a tumor is augmented. One particular immunotherapy, CAR T therapy, involves engineering a patient’s own T cells to express a receptor that specifically targets antigens on tumor cells. This therapy has shown promise in preclinical experiments but has not shown benefit in clinical studies to date. This review explores the potential limitations of this therapy and possible ways of improving its effectiveness in treating glioblastoma.

**Abstract:**

Glioblastoma (GBM) is the most common primary brain tumor, yet prognosis remains dismal with current treatment. Immunotherapeutic strategies have had limited effectiveness to date in GBM, but recent advances hold promise. One such immunotherapeutic advance is chimeric antigen receptor (CAR) T cell therapy, where autologous T cells are extracted and engineered to express a specific receptor against a GBM antigen and are then infused back into the patient. There have been numerous preclinical studies showing promising results, and several of these CAR T cell therapies are being tested in clinical trials for GBM and other brain cancers. While results in tumors such as lymphomas and diffuse intrinsic pontine gliomas have been encouraging, early results in GBM have not shown clinical benefit. Potential reasons for this are the limited number of specific antigens in GBM, their heterogenous expression, and their loss after initiating antigen-specific therapy due to immunoediting. Here, we review the current preclinical and clinical experiences with CAR T cell therapy in GBM and potential strategies to develop more effective CAR T cells for this indication.

## 1. Introduction

Glioblastoma (GBM), histologically defined as grade IV astrocytoma, is a highly malignant brain tumor that affects over 17,000 people in the United States annually [1,2]. GBM carries a particularly poor prognosis, with median survival of 15 months and a five-year survival rate of less than 5% [3]. The standard-of-care therapy for GBM, established by Stupp et al. in 2005, demonstrated improved outcomes for patients treated with maximal safe surgical resection, radiotherapy, and temozolomide, resulting in an average survival of 14.6 months [4]. Given the limited effectiveness of chemotherapy and radiation, alternative treatment strategies are needed that meet the specific challenges imposed by GBM, including early microscopic spread of tumor cells, impaired immunological response, and drug delivery limitations posed by the blood–brain barrier [5,6]. Unfortunately, there has been little clinical progress in the treatment of this disease. Of the phase III clinical trials that have taken place since 2005, only one study has produced positive findings [7]. Immunotherapy is being successfully applied to a growing number of advanced cancers; however, multiple phase III trials of immune checkpoint blockade have failed to meet their endpoints in GBM. Nevertheless, subgroup analyses indicate an immune-based approach may have activity in GBM, a notion that is supported by preclinical data. In this review, we will focus specifically on the potential of CAR T cell therapy, including the data available to date and potential future directions.

Cancer immunotherapy broadly involves activating the immune system to eliminate cancer cells. The most widely studied approaches are immune checkpoint blockade, anti-tumor vaccines, oncolytic viruses, and cellular immunotherapy [8]. Chimeric antigen receptor (CAR)-T cell therapy, a form of cellular immunotherapy, has garnered particular interest in recent years due to its remarkable activity in hematologic malignancies [9,10,11,12]. Generally, CAR T therapy involves extracting T cells from a patient’s blood, engineering these T cells to express synthetic antigen receptors, and infusing these cells to target tumor antigens. While this form of adoptive cell transfer has been performed with T cells harvested directly from tumors, these tumor-infiltrating T cells can be difficult to isolate and expand and may not actually be specific for the tumor [13,14,15]. The synthetic chimeric receptors, on the other hand, include both extracellular domains that recognize tumor-associated antigens (TAAs) and intracellular co-stimulatory signaling domains to promote T cell activation [16]. Initial successes with CAR T therapy have resulted in FDA approval of multiple therapeutics for liquid cancers, and prompted interest in investigating the use of CAR T therapy for solid tumors [17]. 

Recent data demonstrating activity of CAR T cells targeting GAD2 in diffuse intrinsic pontine glioma demonstrate that achieving responses against primary brain tumors with CAR T cell therapy is possible [18]. Here, we review current experiences with CAR T cell therapy in GBM and limitations posed by antigen expression, and discuss strategies to improve CAR T cell efficacy in this context. 

## 2. CAR T Therapy Background

Recent advances in understanding T-cell receptor structure and activation have paved the path for the development of numerous T-cell-based cancer therapies. CAR T-cell therapy is at the forefront of these therapies, currently dominating the clinical trials landscape for cell-based cancer therapy [19]. This therapy involves inducing expression of a chimeric antigen receptor, which is engineered to target a specific antigen of interest, on autologous T cells. Thus, CAR T cell therapy bypasses the requirement for MHC-mediated antigen presentation, which is required for activation of endogenous T cells [20]. This characteristic allows CAR T cells to target a wider array of tumor-associated antigens. In addition to the antigen-specific receptor, CARs contain the intracellular domains necessary for activation so they can perform cytotoxic functions independent of environmental signals. 

Specifically, CAR structure includes extracellular, transmembrane, and intracellular domains [21]. The extracellular domain consists of the antigen-binding domain and a hinge domain. Generally, the MHC-independent binding domain is derived from a single-chain Fragment variant (scFv) of the variable portions of heavy and light chains of a monoclonal immunoglobulin. The transmembrane domain is made of an alpha-helix derived from CD4, CD8α, or CD28. The intracellular domain consists of a T-cell receptor CD3 complex that facilitates signal transduction through activation of downstream kinase pathways, ultimately leading to T-cell activation and cytokine production (Figure 1) [22]. More recent iterations of CAR T cells involve modifying the intracellular domain to improve activation, longevity, and effector function. The first generation of these engineered receptors included a CD3 zeta domain. Notably, they required IL-2 co-stimulation to exert their cytotoxic function. To circumvent this, a second generation of receptors was designed to incorporate co-stimulatory proteins such as CD28, 4–1 BB, or OX40. These engineered cells showed superior antitumor response, with less activation-induced cell death of the CAR T cells, and increased release of pro-inflammatory cytokines [23]. Third-generation CAR T cells integrate multi-stimulatory pathways for T-cell activation, as opposed to a single pathway in the second generation. Finally, fourth-generation CAR T cells are being designed to produce cytokines that activate antitumor immune responses [24]. 

In the last decade, CAR T cell therapy has garnered considerable attention for activity in acute lymphoblastic leukemia (ALL), chronic lymphocytic leukemia (CLL), and B cell lymphomas [19]. Because CD19 is a highly expressed antigen in these hematologic malignancies, it has become the primary target for CAR T therapy in these cancers [25]. The results of this therapy compared with current standard treatments for relapsing B cell lymphoma may soon elevate CAR T cell therapy to standard second-line treatment [26,27]. 

Clinical implementation of these therapies has also highlighted some limitations. For instance, although rare, patients who have recurrence after this therapy often lack CD19 antigen due to immunoediting. Thus, other antigens, such as CD20 or CD22, can be targeted in combination with CD19 to counteract antigen escape and mutation in tumor cells [28]. Additionally, this therapy has been associated with a rare but unique set of side effects. Cytokine release syndrome and neurological toxicity are both potentially life-threatening events resulting from systemic immune activation [29]. Overall, however, these therapies have been safe, with ongoing clinical trials in treatment of ALL, CLL, and non-Hodgkin lymphoma (NHL) continuing to show promising results [30]. Lessons from this work in hematologic malignancies may help inform treatment in solid tumors, including GBM.

## 3. CAR T Therapy in GBM

The success of CAR T therapy in hematologic cancers has driven interest in application of this strategy to GBM. While GBM antigens have been targeted with antibodies/vaccines with modest results, CAR T therapy may have advantages over these therapeutics [31]. For example, there is better blood–brain barrier penetration through utilization of immune cell trafficking [32]. Additionally, this treatment can directly kill tumor cells once bound to receptor, and thus is not dependent on the endogenous immune response, which is severely suppressed in GBM [33,34,35]. However, a major limitation in T cell-based therapies for GBM is the dearth of tumor-specific antigen expression and low mutational burden [36,37]. As the genetic and protein profiles of these tumors continues to be further understood, more potential TAAs have been discovered and used as targets for CAR T cell therapy. In parallel, new approaches are being developed to overcome the challenges of targeting infrequent and intracellular tumor antigens [38]. While many CAR T cells for solid tumors are still in preclinical development, several are now being tested in clinical trials (Table 1).

The most targeted GBM antigen is mutant epidermal growth factor receptor (EGFR). EGFR is a transmembrane receptor tyrosine kinase that can normally be found throughout the body but is minimally expressed in the CNS [39]. EGFR expression is amplified in over 50% of GBMs, making it a promising target [40]. While overexpression of non-mutated EGFR is limited by systemic expression, it often develops mutations late in GBM development [41]. One particular mutation, a deletion of exons encoding the ligand binding region, can be found in 30% of GBMs [39,42]. This mutated receptor variant, called EGFRvIII, is constitutively active and contributes to tumor growth. The clinical relevance of this mutation is supported by the fact that its expression is a negative prognostic factor in patients who survive a year or longer [40]. Because of its highly specific expression on GBM and absence of expression on healthy tissues, EGFRvIII has been explored extensively as a possible target for vaccines, and more recently in CAR T cell therapy [39,43,44]. Multiple clinical trials have been performed with third-generation CAR T cells targeting EGFRvIII [45,46,47]. The most common adverse events seen were related to lymphodepleting chemotherapy, which is given prior to infusion of the CAR T cells [45]. Further, two patients developed severe dyspnea at the highest dose, one of whom died, and several patients developed new minor neurological changes or seizures, thus requiring steroids or antiepileptic medications. No patients were seen to develop severe immunological side effects [46]. Although there are encouraging anecdotes, such as the patient with recurrent GBM who lived for 36 months after EGFRvIII-targeted CAR T cell infusion, reproducible responses have been elusive [47]. The median progression-free survival was merely 1 month, only seen on first-follow up after infusion [45]. The overall median survival after treatment was only 6.9 months.

Another target of interest for GBM is interleukin-13 receptor α chain variant 2 (IL13Rα2). Under normal physiologic conditions, cells express variant 1 of this receptor, which binds IL-13 to activate downstream JAK-STAT signaling, which arrests cell proliferation and drives apoptosis [48]. Variant 2, which lacks the downstream signaling pathway, is restricted to tumor cells (except for the testis), and acts as a decoy receptor that competitively sequesters IL-13, preventing apoptosis of tumor cells and driving tumor progression [49]. IL13Rα2 is expressed on numerous solid tumors, including up to 50–80% of GBM cells [50]. Importantly, there is no overlapping expression in the brain, making it a suitable target for antigen-specific therapies such as CAR T cells. Given that the similar variant IL13Rα1 is broadly expressed throughout the brain, CAR T cells are engineered to exclude cross-reactivity [51]. These therapies have shown success in murine models and have been tested clinically [52]. In a safety trial with three patients with recurrent GBM receiving a first-generation CAR T targeting IL13Rα1, no major adverse events were seen; headaches were attributed to high doses of infusion, and one patient developed neurological changes that improved with steroids [52]. A notable result included a single patient with recurrent multi-focal GBM treated with CAR T cells who initially showed a complete response after infusion, only to have a recurrence 7.5 months later [53]. While this example shows the potential benefit of this therapy, it was not curative, and just as with EGFRvIII CAR T cells, most patients had no significant benefit. On serial imaging, two patients appeared to have decreased expected tumor recurrence in periphery where intracranial injection occurred, and the patients lived 11–14 months after their first recurrence, which was prior to receiving this therapy [52].

Human epidermal growth factor receptor 2 (HER2) has also been utilized as a target for T cell therapy. HER2 is a receptor tyrosine kinase that is expressed on normal epidermal tissue at low levels and overexpressed in several cancers. In GBM, upregulation has been noted in up to 80% of tumors, without any concomitant expression in healthy brain tissue [54]. Unlike EGFRvIII and IL13Rα2, HER2 is expressed at low levels in healthy epidermal tissues, leading to a higher probability of off-target effects. Although most patients have tolerated these infusions well, one patient with colon cancer who received this therapy died from cytokine storm syndrome [55]. Fortunately, no severe adverse events occurred in a clinical trial assessing the safety of a second-generation CAR T targeting HER2 in progressive GBM [56]. As expected, the most common reported adverse events were sequelae of lymphodepletion, and a few patients developed seizures, headaches, or minor neurological deficits. There was possible benefit seen in some patients, as the median survival in 16 patients treated with this therapy was 24.5 months from diagnosis, which was 11.1 month survival after treatment [56]. Three patients showed no progression as late as their 29 month follow-up. Further clinical trials are being conducted.

## 4. CAR T Therapy Evasion in GBM

Despite intensive efforts to find highly specific and robustly expressed antigens in GBM cells, targeting these antigens with CAR T therapy in clinical trials has not yielded significant clinical benefit for patients. One of the major reasons for this failure is antigen escape. Even with the successes seen in hematologic malignancies, with up to 90% remission rates in some cancers, recurrences frequently harbor CD19-negative tumor cells [57,58]. This is believed to be due to selective pressure by the CAR T cells for cancer cells with increasing expression of mutated or truncated variants of CD19, ultimately leading to a subpopulation of cancer cells not detected by the CAR T cells [59]. In a GBM mouse model, even when CAR T therapy was given in conjunction with cytokine stimulation, tumors eventually lost their expression of IL13Rα2, which led to late recurrence [51]. In a phase II clinical trial assessing the immunological effect of EGFRvIII-targeted peptide vaccine, it was shown that the vast majority (82%) of patients who went on to have tumor recurrence had lost EGFRvIII expression [60]. Further, in patients who received CAR T therapy targeting EGFRvIII, there was loss of this antigen expression in five of seven patients who received surgery after infusion, along with increased expression of inhibitory molecules and T regulatory cells [46]. Another patient treated with intraventricular IL13Rα2 CAR T therapy had initial resolution of multifocal GBM, until eventual relapse of disease with tumor cells showing decreased expression of the IL13Rα2 protein [53]. These findings highlight antigen escape as a major challenge for achieving sustained responses to CAR T cell therapy. 

Contributing to antigen escape is the molecular heterogeneity of GBM [61,62]. Not only can antigens have heterogeneous spatial expression throughout a tumor, but they can also vary across time [63]. This variability in antigen expression may be related to a variety of changes in the tumor microenvironment, including immune infiltration, hypoxia, and metabolism [62]. Further, these factors may have different effects on various antigens; for instance, even without the effects of treatment, EGFR expression may be inversely related to that of IL13Rα2 and HER2 [62]. Thus, the antigenic composition of a tumor is influenced by treatment, tumor progression, and environmental factors. Unlike with liquid tumors, immune cells in GBM and other solid tumors may be particularly impacted by these macro and microenvironmental factors; the development of glioblastoma organoids that more accurately portray the heterogeneity of tumors is allowing for better exploration of this complexity, and is only just starting to be used in preclinical studies of CAR T cells [64,65]. Furthermore, immunoediting can occur rapidly and has been documented as soon as 2.5 days in vitro [66]. Thus, the intratumoral heterogeneity of GBM creates two challenges. First, only a subset of cells of the overall tumor may be targeted and, second, the targeted cells may shift their molecular repertoire, leading to further heterogeneity and rapidly dwindling targets. To improve treatment paradigms with this technique, antigen escape and heterogeneity must be taken into consideration.

## 5. Improving CAR T Therapy in GBM

### 5.1. Targeting Alternative Antigens

Ideally, target antigens should be homogenously expressed on tumors without presence in healthy tissue. These characteristics are elusive in GBM. Some newly identified antigens are showing promise in preclinical studies, with a few now being evaluated in new clinical trials. While not an exhaustive list, several antigens are worth discussing (Table 2).

B7-H3 is a transmembrane protein that may have immune checkpoint properties with mixed stimulatory/inhibitory effects on T cells [67]. It is overexpressed in over half of GBM samples and invariably expressed in neurospheres, with minimal expression in normal tissues, making it a promising target [68]. CAR T cells have thus been designed to target this antigen and have shown promising results in preclinical studies. Treatment of U-87, U-138, and GBM neurosphere cell lines has shown elimination of tumor cells in vitro, with induction of interferon gamma and IL-2 production [69]. Further, this CAR T therapy has been shown to control tumor growth when U87 cell lines, patient-derived tumor lines, and neurospheres are implanted orthotopically in murine models, prolonging survival in 50% of these mice compared to control treatment [67,69]. Interestingly, although some of these treated tumors eventually recur, the expression of B7-H3 is retained, which may suggest late recurrence is not due to antigen loss with this treatment. Several clinical trials are currently recruiting patients with recurrent GBM for this therapy.

An antigen found in up to 90% of GBM samples and without expression on surrounding healthy brain is erythropoietin-producing hepatocellular carcinoma A2 (EphA2) [70]. Thought to be involved in oncogenesis, these characteristics make this antigen a potentially useful target in immunotherapy. CAR T cells targeting EphA2 eliminated GBM neurospheres in culture along with preventing their formation [71]. When treating multiple EphA2+ GBM cell lines with optimized CAR T cell design, treatment showed tumor cell lysis with significant upregulation of interferon gamma and IL-2, and only low levels of induced inhibitory cytokines IL-10 and IL-4 [72]. Further, mice implanted with U373 tumor cells had a decrease in tumor burden and improved survival after treatment, with about 50% having complete regression of tumor [71]. Notably, no adverse effects were seen in the treated mice.

CD70 is a transmembrane protein that is the ligand for CD27, an immune cell receptor involved in co-stimulation of lymphocytes via tumor necrosis factor pathway [73]. Expression of CD70 has been seen mostly in mesenchymal GBM cell subtypes and is negatively associated with survival. CAR T therapy targeting this molecule on tumor cells causes tumor cell death and increases expression of IFN gamma and TNF alpha in culture supernatant [74]. Additionally, mice with implanted GBM cell lines had a survival advantage when treated with CD70 CAR T cells; complete resolution was seen in tumors derived from CD70+ clones, whereas improved survival with 38% cure was seen in tumors with heterogeneous (and thus more representative) CD70 expression. These results were seen in both xenograft and syngeneic murine models, without any significant adverse effects [73]. One concern in targeting this protein is the potential to induce immunosuppression given its role in normal T cell activation; however, preclinical evidence does not suggest this effect is significant, with no inhibition of the adaptive immune response seen one month after administration [75]. By further modifying these CAR T cells with an additional receptor that targets IL-8 secreted by tumor cells, there was improved tumor penetrance and persistence by these CAR T cells, leading to complete tumor regression in a U87 murine model [76]. This modified CD70 CAR T cell will be assessed in a clinical trial that is being initiated soon to assess safety in GBM patients.

Natural killer (NK) cells are one of several cytotoxic immune cells. They perform surveillance under normal conditions and become activated when their receptor, natural killer group 2-member D (NKG2D), binds various ligands expressed on cells during cellular stress [77]. Tumors including GBM are known to express NKG2D ligands; however, the immunosuppressive microenvironment reduces the levels and efficacy of the receptor on NK cells, rendering them unable to target GBM [78]. A unique and attractive quality of this receptor for CAR therapy is that it has several different stress-related ligands it binds to [79]. This has a potential advantage over other single-antigen CAR T cells in preventing antigen escape. When engineered into CAR T cells, NKG2D-targeted therapy showed cytotoxicity toward T98, U251, and U87 GBM cell lines in vitro, measured by increased cytokine production as well as granzyme B and perforin [80]. There was also activity noted against GBM neurospheres in vitro, and in murine orthotopic models, mice treated with NKG2D CAR T cells showed complete elimination of tumors without recurrence 42 days after treatment. These CAR T cells preferentially accumulated within the tumors in the mice, with little infiltration in and no noted immune effect on systemic organs. With this positive safety profile, several clinical trials assessing the safety and efficacy of this specific CAR T therapy in solid tumors will also assess its effect in GBM patients.

Seen widely expressed in various solid tumors, GD2 is a disialoganglioside being tested as a target for CNS tumors, particularly childhood tumors [18,81]. This carbohydrate-containing sphingolipid has been implicated in tumor development through various means [82]. After confirmation that it is indeed expressed in GBM, mice were implanted with human tumors and then treated both intravenously and intracranially with CAR T cells targeting GD2 [83]. The latter treatment group saw a robust increase in survival, while in vitro models showed increased cytokine release with treatment. Another study showed survival benefit with peripheral injection, and further improvement in tumor control with co-expression of transgenic IL-15 [84]. Researchers in China have further tested a CAR T cell targeting this antigen in eight patients with GBM, which caused no severe adverse events and reduced tumor size in half of the patients after infusion; however, ultimately there was no survival benefit [85].

Chondroitin sulfate proteoglycan 4 (CSPG4), also known as Neuroglia-2 (NG2), is a protein that is expressed in 67% of GBM specimens and in a more homogenous manner than other antigens [86]. It has been shown to be involved in promoting tumor progression and inducing chemotherapy and radiotherapy resistance [87,88]. CAR T therapy targeting CSPG4 in GBM neurospheres showed cytotoxic effects against tumor cells, induced IFN-γ and IL-2, and showed proliferation in the presence of tumor cells [86]. Further, multiple murine models of different GBM and neurosphere cell lines treated by intratumor injection showed significant tumor control, with prolonged survival in 42–97% of treated mice depending on cell line. A unique characteristic of this antigen is that it is inducible by TNF-α secretion from microglia, which may allow targeting of this antigen with CAR T cells even when little expression of CSPG4 is seen to begin with, as the immune cascade eventually yields its expression on GBM cells [86]. These findings make targeting CSPG4 with CAR T cell technology a promising therapy that needs further study.

Another potential target is chlorotoxin (CLTX), a peptide isolated from the venom of the death stalker scorpion. This molecule has previously has been shown to selectively bind to tumors of neuroectodermal origin, including gliomas, without binding to normal brain [89]. In fact, surgeons have utilized this property by intraoperative use of fluorescently labeled CLTX to identify tumor from healthy tissue [90]. This effect is due to binding of the toxin with matrix metalloproteinase-2 (MMP-2), a membrane-bound protein widely expressed in GBM [91]. When engineered as a receptor for CAR T cells, this therapy showed broad treatment effect in a murine GBM model, and particularly notable was its effect in tumors not expressing other frequently targetable antigens [91]. Treatment of tumors implanted in mouse flanks yielded complete tumor regression, while treatment of intracranial implanted tumors yielded about 50% survival. It was noted that different GBM cell lines responded differently to this treatment, and those with less response were seen to have upregulated immune checkpoints, suggesting exhaustion as the main driver in recurrent cases. This CAR T therapy is being investigated in a phase I clinical trial that is actively recruiting patients.

### 5.2. Targeting Multiple Antigens

Targeting multiple antigens is another strategy to combat antigen escape, as tumor cells may be able to downregulate a single antigen without affecting critical functions but may not be able to simultaneously alter multiple pathways. A second advantage of this approach for heterogenous tumors such as GBM is that a higher percentage of tumor cells will be targeted, leaving fewer clones capable of escape. To meet this need, CAR T cells are being developed with either multiple different receptors, tandem receptors, or a single CAR with multiple antigen-binding domains. Much of this work has been done in hematologic malignancies. For instance, many patients with aggressive B cell lymphoma who are treated with CAR T therapy targeting CD19 may relapse, with 30% of those being CD19-negative [92]. CAR T cell therapy targeting CD20 in combination with CD19 showed superior clinical results, with fewer CD19-negative recurrences [28]. Similarly, in a GBM tumor model of kinase inhibitor resistance, dual therapy prevented the proliferation of subpopulations of cells not targeted with monotherapy [66]. T cells bearing combination HER2 and IL13Rα2 CARs showed benefit both in vitro and in a murine glioma model [93]. Not only was antigen loss absent, but there were also increased antitumor effects associated with use of the dual-targeted CAR T cell approach. Combining CD70 and B7-H3 in tandem showed preclinical evidence of improved anti-tumor response in multiple cancers, including glioma [94]. Triple-antigen primed CAR T therapy is also being used. One trivalent therapy was developed to target HER2, IL13Rα2, and EphA2 in GBM. This trivalent approach showed superior benefit to single or dual antigen approaches, and predictive modeling suggested the ability to kill nearly all tumor cells from 15 different patients despite their antigen expression variability [95]. 

Given the inherent limitation of tumor specific antigens without off-tumor cross reactivity and the heterogeneous nature of antigen expression on GBM, another strategy has been developed with a so-called prime-and-kill circuit. A synthetic Notch receptor can be engineered to recognize a certain antigen, leading to transcription of a CAR [96]. Choe et al. used this technique to develop a synNotch CAR T cell against GBM, whereby first the synNotch receptor recognizes either a GBM-specific antigen (EGFRvIII) or CNS-specific antigen (MOG), priming the cell to then express a tandem CAR targeting GBM antigens, EphA2 and IL13Rα2 [97]. In this way, it will only become active in the vicinity of the tumor, and then target various other antigens without concern for off-tumor cytotoxicity. This strategy has potential to subvert antigen loss associated with solely targeting single, highly specific TAAs, while also preventing adverse effects on healthy tissues that may be associated with less-specific multi-valent strategies. What is also interesting about this approach is that the cells can be primed by CNS-specific, rather than tumor-specific, antigens. This is highly applicable to GBM where there is limited highly specific TAA.

### 5.3. Targeting Antigens of Cancer Stem Cells

While initial efforts with CAR T therapy for GBM targeted surface antigens on differentiated tumor cells, there is appeal in targeting cancer stem cells to limit growth and recurrence. It is thought that within a tumor, there are only a few cells that possess the ability to propagate uninhibited and lead to differentiated cancer cells [98]. Without eliminating these precursor cells, tumors will recur even if the bulk of tumor cells are eliminated [99]. Standard chemoradiation has limited activity against GBM stem cells [100,101]. Thus, developing a therapy with activity against these cancer stem cells is critical, and CAR T therapy shows promise. Work with previously developed CAR T cells has demonstrated antitumor activity against GBM stem cells. For instance, expression of IL13Rα2 has been noted in some GBM stem cells, defined by CD133 expression, with their selective killing by CAR T cells targeting this molecule [102]. However, its heterogeneous expression within this cell population does not allow for complete eradication of the multipotent tumor cells. Similarly, therapy targeting HER2 has shown efficacy in eliminating HER2/CD133 positive GBM stem cells but leaves the HER2-negative subpopulation untouched [54]. Targeting of stem cell markers together with other TAAs may specifically eliminate these stem cells. The expression of EGFRvIII can be found on a subset of tumor cells expressing CD133 [103]. A chimeric antibody with bispecific affinity for both EGFRvIII and CD133 showed increased efficacy in cytotoxicity and decreased self-renewal abilities of GBM cells and implanted tumors. Even more promising, B7-H3 is highly expressed in patient-derived GBM neurospheres, a cell population enriched in cancer stem cells and possibly more representative of primary GBM [69]. CAR T cells targeting this antigen showed antitumor activity for both neurospheres as well as established GBM cell lines, suggesting antitumor effects against both stem cells and differentiated cells. Although still being explored, eliminating GBM stem cells has important implications for potential curative combinations.

### 5.4. Adjuvant Therapies to Increase Antigen Availability and CAR T Efficacy

While improving CAR T cell design may increase efficacy against the limited number of antigens present on the tumor, inducing antigen expression may be another strategy to improve CAR T therapy results. Radiation therapy, which is part of the standard-of-care treatment for GBM, may have a beneficial role in CAR T cell therapy. For instance, irradiation of tumor cells may induce further expression of TAAs. GBM cell lines with CD70 positivity showed increased expression of this antigen after exposure to irradiation [73]. Similarly, NKG2D has been shown to increase expression on GBM following irradiation; mice bearing glioma tumors had improved survival with combination irradiation and CAR T therapy targeting NKG2D, compared to monotherapy alone [104]. This was shown to be due to two mechanisms, one being the direct induction of the antigen itself after irradiation, and another due to increased CAR T cell migration to the tumor. Interestingly, as seen in a pancreatic tumor model, low-dose radiation may also prime CAR T cells to attack tumor in a non-antigen-dependent manner [105]. This effect could enhance CAR T cell efficacy even if there is escape of the primary targeted TAA. Thus, radiation may have multiple synergistic effects with CAR T therapy when performed in a coordinated manner. Similar results have been shown with chemotherapy. Chemotherapy can induce the expression of tumor antigens in solid tumors, leading to a more robust immune response [106]. This has been shown with NKG2D CAR T cells in GBM models [79]. These findings are especially relevant given that CAR T therapies for GBM will likely be tested in patients receiving chemotherapy and radiation.

Oncolytic viruses, which can specifically infect tumor cells and have well-documented immune-mediated activity, can be used in conjunction with CAR T therapy. Their direct lytic effect on tumor cells not only can trigger inflammation but can also cause the tumor cells to release TAAs [107]. Additionally, oncolytic viruses can also be used to induce expression of any desired de novo antigen by tumor cells; for instance, a virus was designed to induce expression of CD19 on solid tumor cells that otherwise would not express this protein, allowing for successful treatment with a CAR T cell targeting CD19 [108]. In tumors that have few and heterogenous antigens, this could have significant benefits. 

Focused ultrasound is another emerging technology with potential immunotherapeutic potential in the treatment of brain tumors. Direct tumor ablation with acoustic energy can release tumor debris containing neoantigens, of which targeted CAR T therapy can take advantage [109]. Additionally, focused ultrasound can also help with drug delivery, not only via blood–brain barrier disruption but also through acoustogenetic engineering whereby the CAR T cell is only activated after exposure to acoustic energy [110]. Further work is needed to better understand the synergistic effects of these adjuvants with CAR T cells.

### 5.5. CAR T Cells and Immune Exhaustion

Despite best efforts to optimize CAR T cell design, immune exhaustion is another limiting step with this therapy. Briefly, the immune system has built-in checks and balances to prevent immunotoxicity when fighting chronic infection [111]. When T cells have prolonged exposure to an antigen, they can become exhausted, having decreased effect or function and increased inhibitory receptors called checkpoints. Cancer, including GBM, takes advantage of this by secreting immunoinhibitory signals to dampen the inherent immune system, which can also decrease the efficacy of CAR T cells [112]. Thus, addressing exhaustion may improve the overall efficacy of CAR T therapy. 

Inhibiting immune checkpoints that are upregulated in GBM continues to be extensively explored as a way of reversing the exhaustive immune state in the tumor microenvironment, with the goal of improving native anti-tumor immune response [113]. Combining checkpoint inhibitors with CAR T therapy may thus improve overall CAR T cell effect and has been assessed in preclinical studies [114]. For instance, the blockade of immune checkpoints (PD-1, CTLA-4, and TIM-3), in addition to CAR T cell therapy, showed improved efficacy of the treatment in a D270 murine model, as seen by tumor growth control and survival [115]. Which checkpoint inhibitor improved therapy the most was dependent on which CAR T cell was used, as CAR T cells targeting EGFRvIII and IL13Rα2 both induced different checkpoint milieus in their respective tumor microenvironments. Clinical trials are starting to look into the safety of this dual therapy in patients with GBM, using CAR T cells against EGFRvIII along with Pembrolizumab (NCT03726515) and CAR T cells targeting IL13Rα2 along with Nivolumab or Ipilimumab (NCT04003649). Additionally, CAR T cells can be engineered to express proinflammatory molecules. CAR T cells that secrete checkpoint inhibitors are not only as efficacious in slowing GBM growth in murine model as co-administration of the checkpoint inhibitor, but they also have more specific targeting, as seen by increased IFN gamma and TNF alpha secretion. In another study, the proliferative capacity and persistence of CAR T cells against IL13Rα2 in a murine model was improved with concomitant IL-15 transgenic expression [51]. 

Another way to address exhaustion in CAR T therapy is through resting CAR T cells via transient downregulation of the receptor [116]. As T cell exhaustion is driven by chronic antigen exposure, temporarily preventing CAR signaling may block epigenetic changes that lead to exhaustion. By designing CARs with drug-dependent destabilizing regions, the expression of the CAR can be regulated and thus transiently shut off. This cessation of CAR expression induced CAR T-cell progression toward a memory rather than exhausted state. Even more, CAR T cells already in an exhausted phenotype showed epigenetic reprogramming with function reinvigoration [116]. Notably, this is not seen with checkpoint inhibition. Thus, addressing the exhaustive pressure the tumor microenvironment places on CAR T cells is one more way of hopefully improving the efficacy of this treatment modality in GBM.

## 6. Conclusions

CAR T cell therapy is a promising tumor immunotherapy that has characteristics uniquely suited to the challenges posed by GBM. These cells can traffic into the CNS and eliminate tumor cells with precision and minimize collateral damage. Next-generation CAR T cells may also deliver payloads that increase the immunogenicity of the GBM microenvironment. To date, promising preclinical findings have not translated to clinical trials. These failures, however, are elucidating the obstacles hindering success, and new strategies are being developed in response. Antigen escape is being addressed by CAR T cells with multivalent receptors, induced antigens, and combination therapies. Adjuvants with other immunotherapies, chemotherapies, or mechanically ablative methods may improve results as well by creating a more favorable, pro-inflammatory microenvironment. Continued work to better understand how the tumor, tumor microenvironment, and host immune system interact with CAR T therapy will inform further CAR T cell development and bring us closer to an effective immunotherapy for GBM.

## Figures and Tables

**Figure 1 cancers-15-01414-f001:**
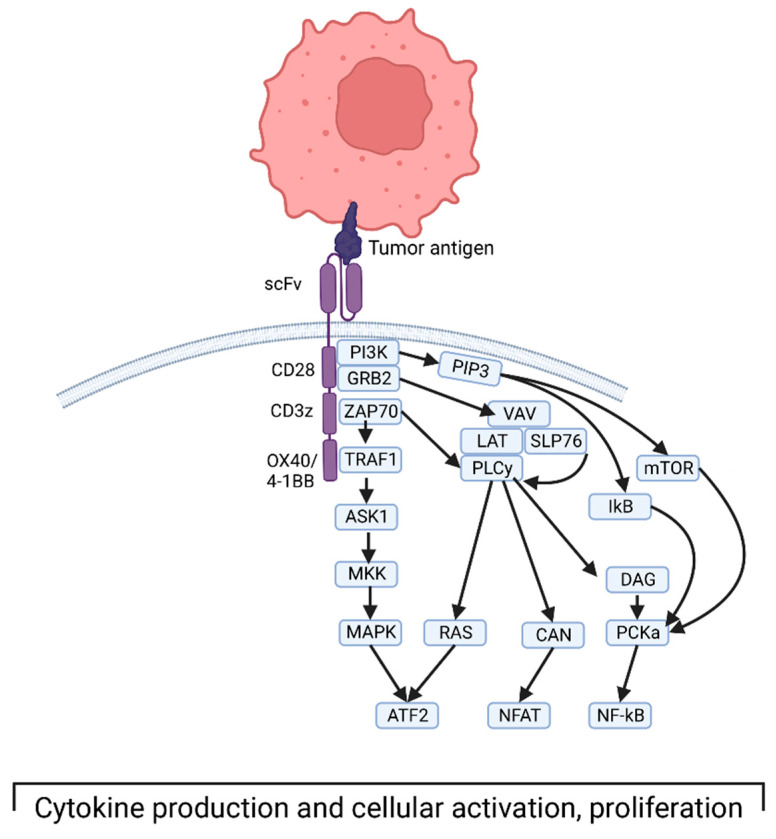
Depiction of third-generation CAR T cell interacting with tumor antigen and the downstream intracellular signaling that ultimately leads to T-cell activation and cytokine production. Created with BioRender.com, accessed on 17 December 2022.

**Table 1 cancers-15-01414-t001:** Current list of clinical trials using CAR T therapy in adult patients with GBM, including those that have been completed and those that are actively or soon to be recruiting patients. Information obtained from ClinicalTrials.gov, accessed on 5 January 2023.

Status	Target Antigen	Title	Primary Site	ID
Completed	EGFRvIII	CART-EGFRvIII + Pembrolizumab in GBM	Abramson Cancer Center of the University of Pennsylvania, Philadelphia, PA	NCT03726515
Completed	EGFRvIII	CAR T Cell Receptor Immunotherapy Targeting EGFRvIII for Patients with Malignant Gliomas Expressing EGFRvIII	National Institutes of Health Clinical Center, 9000 Rockville Pike, Bethesda, MD	NCT01454596
Completed	IL13Rα2	Cellular Adoptive Immunotherapy Using Genetically Modified T-Lymphocytes in Treating Patients with Recurrent or Refractory High-Grade Malignant Glioma	City of Hope Medical Center, Duarte, CA	NCT00730613
Completed	HER2	CMV-specific Cytotoxic T Lymphocytes Expressing CAR Targeting HER2 in Patients With GBM	Houston Methodist Hospital, Houston, TX; Texas Children’s Hospital, Houston, TX	NCT01109095
Active, not recruiting	EGFRvIII	The Efficacy and Safety of Brain-targeting Immune Cells (EGFRvIII-CAR T Cells) in Treating Patients with Leptomeningeal Disease from Glioblastoma. Administering Patients EGFRvIII -CAR T Cells May Help to Recognize and Destroy Brain Tumor Cells in Patients	Jyväskylä Central Hospital, Jyväskylä, Finland; University of Oulu, Oulu, Finland; Apollo Hospital, New Delhi, India	NCT05063682
Active, not recruiting	IL13Rα2	Genetically Modified T-cells in Treating Patients with Recurrent or Refractory Malignant Glioma	City of Hope Comprehensive Cancer Center, Duarte, CA	NCT02208362
Recruiting	B7-H3	Safety and Efficacy Study of Anti-B7-H3 CAR-T Cell Therapy for Recurrent Glioblastoma	Beijing Tiantan Hospital, Beijing, China	NCT05241392
Recruiting	B7-H3	B7-H3 CAR-T for Recurrent or Refractory Glioblastoma	Second Affiliated Hospital of Zhejiang University School of Medicine, Hangzhou, Zhejiang, China; Huzhou Central Hospital, Huzhou, Zhejiang, China; Ningbo Yinzhou People’s Hospital, Ningbo, Zhejiang, China	NCT04077866
Recruiting	B7-H3	Autologous CAR-T Cells Targeting B7-H3 in Recurrent or Refractory GBM CAR.B7-H3Tc	Lineberger Comprehensive Cancer Center, Chapel Hill, NC	NCT05366179
Recruiting	B7-H3	B7-H3 Chimeric Antigen Receptor T Cells (B7-H3CART) in Recurrent Glioblastoma Multiforme	Stanford Cancer Institute, Palo Alto, CA	NCT05474378
Recruiting	Chlorotoxin	Chimeric Antigen Receptor (CAR) T Cells with a Chlorotoxin Tumor-Targeting Domain for the Treatment of MMP2+ Recurrent or Progressive Glioblastoma	City of Hope Medical Center, Duarte, CA	NCT04214392
Recruiting	HER2	Memory-Enriched T Cells in Treating Patients with Recurrent or Refractory Grade III-IV Glioma	City of Hope Medical Center, Duarte, CA	NCT03389230
Recruiting	EGFRvIII	Autologous CAR-T/TCR-T Cell Immunotherapy for Malignancies	The First Affiliated Hospital of Zhengzhou University, Zhengzhou, Henan, China	NCT03638206
Recruiting	EGFRvIII	Autologous CAR-T/TCR-T Cell Immunotherapy for Solid Malignancies	Henan Provincial People’s Hospital, Zhengzhou, Henan, China	NCT03941626
Recruiting	EGFRvIII, GD2, IL13Rα2, Her2, EphA2, CD133	Personalized Chimeric Antigen Receptor T Cell Immunotherapy for Patients with Recurrent Malignant Gliomas	Xuanwu Hospital, Beijing, China	NCT03423992
Recruiting	IL13Rα2	A Clinical Study of IL13RŒ ± 2 Targeted CAR-T in Patients with Malignant Glioma (MAGIC-I)	National Cancer Center, Korea, Goyang-si, Gyeonggi, Republic of Korea	NCT05540873
Recruiting	IL13Rα2	IL13Rα2-CAR T Cells with or Without Nivolumab and Ipilimumab in Treating Patients With GBM	City of Hope Medical Center, Duarte, CA	NCT04003649
Recruiting	IL13Rα2	Brain Tumor-Specific Immune Cells (IL13Ralpha2-CAR T Cells) for the Treatment of Leptomeningeal Glioblastoma, Ependymoma, or Medulloblastoma	City of Hope Medical Center, Duarte, CA	NCT04661384
Recruiting	NKG2D	NKG2D-based CAR T-cells Immunotherapy for Patient With r/r NKG2DL+ Solid Tumors	Xunyang Changchun Shihua Hospital, Jiujiang, Jiangxi, China	NCT05131763
Not yet recruiting	CLTX	A Phase 1 Study to Evaluate CHM-1101 CAR T Cells in Patients with MMP2+ Recurrent or Progressive Glioblastoma	City of Hope Medical Center, Duarte, CA	NCT05627323
Not yet recruiting	Dual CD70, IL8 (CXCR2)	Phase I Study of IL-8 Receptor-modified CD70 CAR T Cell Therapy in CD70+ and MGMT-unmethylated Adult Glioblastoma (IMPACT)	University of Florida Health, Gainesville, FL	NCT05353530
Not yet recruiting	EGFRvIII	Long-term Follow-up of Subjects Treated with CARv3-TEAM-E T Cells	Massachusetts General Hospital, Boston, MA	NCT05024175
Not yet recruiting	IL7R1 (CD127)	Tris-CAR-T Cell Therapy for Recurrent Glioblastoma	Beijing Tiantan Hospital, Beijing, China	NCT05577091
Not yet recruiting	NKG2D	Pilot Study of NKG2D CAR-T in Treating Patients with Recurrent Glioblastoma	Not listed	NCT04717999
Not yet recruiting	NKG2D	NKG2D CAR-T(KD-025) in the Treatment of Relapsed or Refractory NKG2DL+ Tumors	The Affiliated Nanjing Drum Tower Hospital of Nanjing University Medical School, Nanjing, Jiangsu, China	NCT04550663

**Table 2 cancers-15-01414-t002:** List of most studied antigens used for CAR T therapy in GBM, and status of use in clinical trials.

Antigen	Full Name	Description	Clinical Trial Status
EGFRvIII	Epidermal growth factor receptor, variant 3	Transmembrane receptor tyrosine kinase that is amplified in half of GBM tumors, minimally expressed in CNS	First trials completed, ongoing recruitment in multiple other trials
IL13Rα2	Interleukin-13 receptor α chain variant 2	Receptor that binds IL-13 to activate downstream JAK-STAT signaling to promote apoptosis; acts as decoy receptor in GBM	First trial completed, ongoing recruitment in multiple other trials
HER2	Human epidermal growth factor receptor 2	Receptor tyrosine kinase normally expressed in epidermal tissue at low levels, upregulated in 80% of GBM tumors	First trial completed, ongoing recruitment in multiple other trials
B7-H3	B7 homolog 3 protein	Transmembrane immune checkpoint protein with mixed stimulatory/inhibitory properties, expressed on over half of GBM tumors	Active recruitment in multiple trials
EphA2	Erythropoietin-producing hepatocellular carcinoma A2	Protein involved in oncogenesis, expressed in over 90% of GBM but not healthy CNS tissue	Active recruitment in single trial
CD70	Cluster of differentiation 70	Transmembrane ligand of CD27, a co-stimulatory immune cell receptor that activates TNF pathway, expressed in mesenchymal GBM cells	Not yet recruiting
NKG2D	Natural killer group 2-member D	Binds various ligands expressed during cellular stress, including ligands expressed by GBM tumor cells	Active recruitment in single trial
GD2	Disialoganglioside	Carbohydrate-containing sphingolipid seen involved in tumor development, expressed in GBM	Active recruitment in single trial
CSPG4	Chondroitin sulfate proteoglycan 4	Protein homogeneously expressed in 67% of GBM, involved in tumor progression and chemotherapy/radiotherapy resistance	No clinical trial submitted to date
CLTX	Chlorotoxin	Peptide isolated from death stalker scorpion, selectively binds to GBM without binding to normal brain	Active recruitment in single trial

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
