# Peer review of "CAR T Cell Therapy in Glioblastoma: Overcoming Challenges Related to Antigen Expression"

_cancers, 2023, doi:10.3390/cancers15051414_

Round 1
Reviewer 1 Report
The review by Luksik et al on CAR-T cell therapy in glioblastoma appears to be very well-written, comprehensive, and informative.
I have a few suggestions. The authors are free to address them or not. However, addressing them could elevate the impact of the article.
1. Create a Table like Table1 summarizing status of CAR-T cell therapy on non-GBM solid tumors (include ‘outcome’ and ‘side effect’ columns. Also indicate which generation of CAR-T cells were used in each trial.
2. Create a Table summarizing status of CAR-T cell therapy on all tumors (this is probably going to be a large Table; in which case the Table could be provided as a supplementary material). Also indicate which generation of CAR-T cells were used in each trial.
3. Consider a discussion on the status of CAR-T cell therapy in GBM in the context of GBM-organoids.
4. The fact that CAR-T cell therapy remained essentially unhelpful for treating solid tumors may imply that the engineered T-cells are not evenly/adequately reaching all the tumor cells at the same time (or within a short period of time). Consider discussing this possibility. (GBM-organoids and/or xenograft models may be employed to address this issue).
5. I am assuming that for all CAR-T cell-based therapies, the T-cells are harvested from blood. I wonder if it would make sense to harvest tumor-resident T-cells (if available) for creating CAR-T cells against glioma/solid tumors. I would like the authors discuss this premise.
Author Response
Point 1. Create a Table like Table1 summarizing status of CAR-T cell therapy on non-GBM solid tumors (include ‘outcome’ and ‘side effect’ columns. Also indicate which generation of CAR-T cells were used in each trial.
Response 1. While including table summarizing status of CAR-T therapy in other tumors would certainly be more encompassing of the current state of this therapy, because the topic is “Treatment of Glioma”, we feel this would be outside the scope of this review. Discussion of the outcome, side effects, and CAR T generation used in the few GBM clinical trials has been expanded upon.
Point 2. Create a Table summarizing status of CAR-T cell therapy on all tumors (this is probably going to be a large Table; in which case the Table could be provided as a supplementary material). Also indicate which generation of CAR-T cells were used in each trial.
Response 2. Please see response 1.
Point 3. Consider a discussion on the status of CAR-T cell therapy in GBM in the context of GBM-organoids.
Response 3. GBM-organoids are an interesting area of research which will allow for testing CAR-T therapy in vitro while preserving heterogeneity of the tumor. We added brief discussion of this in the tumor heterogeneity paragraph.
Point 4. The fact that CAR-T cell therapy remained essentially unhelpful for treating solid tumors may imply that the engineered T-cells are not evenly/adequately reaching all the tumor cells at the same time (or within a short period of time). Consider discussing this possibility. (GBM-organoids and/or xenograft models may be employed to address this issue).
Response 4. Along with response 2, this is addressed in the tumor heterogeneity paragraph.
Point 5. I am assuming that for all CAR-T cell-based therapies, the T-cells are harvested from blood. I wonder if it would make sense to harvest tumor-resident T-cells (if available) for creating CAR-T cells against glioma/solid tumors. I would like the authors discuss this premise.
Response 5. This is an excellent point, and brief mention of this with references was added to the introduction. Tumor infiltrating T cells are not ideal given they are difficult to isolate and expand, and they lack specificity for the tumors.
Reviewer 2 Report
The review focused on the current preclinical and clinical experiences with CAR T cell 24 therapy in GBM, and potential strategies to develop more effective CAR T cells for this indication. This review is of value and interest, and a minor revision is recommended prior to publication.
1. It is recommended that the targeted antigens be summarized in a table.
2. It is recommended that the results and adverse reactions of CAR-T clinical trials be described in detail.
3. It is recommended to update the following literature:PMID36261831, PMID: 36617554.
Author Response
Point 1. It is recommended that the targeted antigens be summarized in a table.
Response 1. Thank you for your suggestion, this table was added (Table 2).
Point 2. It is recommended that the results and adverse reactions of CAR-T clinical trials be described in detail.
Response 2. Thank you for your suggestion, we expanded the discussion of the results and adverse reactions of the clinical trials that were discussed.
Point 3. It is recommended to update the following literature:PMID36261831, PMID: 36617554.
Response 3. Thank you for your suggestion, we added these references along with an additional discussion of the antigen, GD2.